# Is Reduced Visual Processing the Price of Language?

**DOI:** 10.3390/brainsci12060771

**Published:** 2022-06-12

**Authors:** Christer Johansson, Per Olav Folgerø

**Affiliations:** Department of Linguistic, Literary, and Aesthetic Studies, University of Bergen, 5007 Bergen, Norway

**Keywords:** language origins, language evolution, perception, vision, cognition, recidation, self-domestication, cave art

## Abstract

We suggest a later timeline for full language capabilities in Homo sapiens, placing the emergence of language over 200,000 years after the emergence of our species. The late Paleolithic period saw several significant changes. Homo sapiens became more gracile and gradually lost significant brain volumes. Detailed realistic cave paintings disappeared completely, and iconic/symbolic ones appeared at other sites. This may indicate a shift in perceptual abilities, away from an accurate perception of the present. Language in modern humans interact with vision. One example is the McGurk effect. Studies show that artistic abilities may improve when language-related brain areas are damaged or temporarily knocked out. Language relies on many pre-existing non-linguistic functions. We suggest that an overwhelming flow of perceptual information, vision, in particular, was an obstacle to language, as is sometimes implied in autism with relative language impairment. We systematically review the recent research literature investigating the relationship between language and perception. We see homologues of language-relevant brain functions predating language. Recent findings show brain lateralization for communicative gestures in other primates without language, supporting the idea that a language-ready brain may be overwhelmed by raw perception, thus blocking overt language from evolving. We find support in converging evidence for a change in neural organization away from raw perception, thus pushing the emergence of language closer in time. A recent origin of language makes it possible to investigate the genetic origins of language.

## 1. Introduction

A recent article [1] argued that superior visual perception was necessary for the creation of Paleolithic cave paintings because of the level of correct anatomical details and accurate depictions of high-speed leg positions of animals in motion, considering that the works were accomplished far removed from the actual animals and with crude tools. The article uncovered and outlined current evidence for an association between visual thinkers (some diagnosed within the Autism Spectrum Disorder) and a relatively high percentage of archaic genes, of which some are associated with perception and cognition. Moreover, within this group are some savants who can quickly and accurately scan what they see and reproduce it artistically in extraordinary detail. One example is reproducing the correct number and relative size of windows from a brief exposure to a city scene. However, the linguistic abilities of visual thinkers may be impaired, which suggests a negative correlation between visual perception/memory and language. Possibly, this negative correlation involves an impaired propensity for symbolic thinking: noted in drawing detailed and accurate images as opposed to the iconic stick figures made by normally developing children, who instinctively forget about adding the details in their drawings while telling vivid stories about their drawings. It is difficult to know to what extent thinking is symbolic in the normal population and much more so in the autistic population. An ability to visualize a problem in great detail can often be very useful, for example, for innovators and scientists, where an extremely verbal, symbolic thinker might not see the concrete details. However, there are visual thinkers that can report on their problems with symbolic thinking and their delays in language development (see Section 1.1).

The present article further develops a model of the relation between the visual, the auditive, and the linguistic components in an evolutionary perspective. Compared to cave painters, modern humans have generally lost the extreme ability to remember visual details and reproduce them artistically [1], but the natural ability is still present in visual thinkers and autistic savants [1] and can be greatly improved with technical aids (e.g., photographs, video, light projection) and schooling. There is also contemporary evidence for pre-human superior abilities to remember symbolic/iconic sequences after extremely short exposures in chimpanzees [2]. Pre-human primates, more generally, have analogues to our language processing areas, although these analogues (e.g., the Broca homologue, the Planum Temporale, the Arcuate Fasciculus) are in, or near, the pathways connecting visual areas to prefrontal areas. Thus, these areas will have to deal with much more information than if better separated from the flood of information from the visual areas. What modern humans have gained, however, is the ability of linguistic communication and human speech. This involves a decoupling of cognition, perception, and sensory information. We, therefore, raise the question: Is reduced visual processing the price we had to pay for the evolution of language? First, we will review what is currently known about the evolution of the visual areas and speech areas.

### 1.1. High Visual Acuity in the Visual Thinkers Now and Then

Some modern-day people within the autism spectrum (ASD), particularly the savants, show a huge discrepancy between visual (memory) talent and their communication skills, often marred by delayed or impaired language and impaired social skills in general. One example is Nadia, who drew horses with great artistic talent at the age of four [3,4] while exhibiting severe communication impairment. Is there an analogy between superior visual aptitude in modern savants and that of paleolithic cave painters?

Some people are visual thinkers, and many of them show amazing talent. Professor Temple Grandin gives a unique introduction to the life of an autist in her autobiographical book [5]. She characterizes herself as a primarily visual thinker, using her extraordinary abilities to overcome many obstacles that face autistic people. She claims to convert symbolic verbal information into mental images, and she has problems with abstract states that cannot be easily visualized, such as mapping “happy” to an image of eating toast, if that made her happy at that time. The difficulty with symbolic thinking does not mean that abstract thinking is impossible; it just takes more creativity to come up with vivid imagery [5]. Grandin remembers that as a child, she often left out the function words (such as determiners, pronouns, and copula verbs) as they had no meaning to her [5]. 

Associative thought patterns may override logical thought, such as not being afraid of heights is associated with an airplane’s ability to fly high [5]. Access to detailed imagery makes generalization harder to accomplish because of the high specificity of an image compared to general symbols. In a detailed image, it is not obvious what is more important, whereas verbal symbols may facilitate a focus on one feature at a time. Even though an overt symbol in speech might have many interpretations in different contexts, most people effortlessly focus on only one aspect of a word or construction at a time, forgetting about ambiguities that are not relevant in the context. Ambiguity and idiomatic expressions are problematic for visual thinkers within the autism spectrum disorder, as the expressions have more than one meaning and the literal meaning is not the intended meaning. An image is much more explicit in its representation—it presents what it is. The high specificity of an image makes such representations information-heavy, which implies an elevated magnitude of neural activation to capture every detail. 

Research indicates that children with autism can be overly sensitive to perceptual stimuli because their brains are hyper-connected [6] (inter al.). Grandin also states that with ASD, there are many extremes; some people are extremely visual thinkers, others are pattern-oriented and drawn to music and mathematics, and others are extremely verbal [5]. In the non-autistic population, we find all these thought patterns in a more balanced distribution, where few are in the extremes.

One pertinent question is if extreme visual thinking may be caused by an imbalance in the reciprocal pathways between first-order perception in the primary visual areas (V1/V2) feeding bottom-up and cortical areas feeding top-down conceptual information back to those same visual areas? One answer to this question is given by Hutsler and Casanova: “(f)unctional imaging studies have shown a reduction in coordinated activity between distant cortical regions and an absence of the top-down modulation of early sensory processing that is found in typically developing individuals” [7] (p. 123). This balance of excitation and inhibition of target cells is mostly modulated by neurotransmitters glutamate and GABA, respectively.

It is obviously difficult to know the differences in the neuro-signal modulation between the two homo species (sapiens sapiens and sapiens neanderthalensis), but more details on the genome of both emerge every year. Genes found in archaic sapiens (including neanderthalensis) such as CUTA and PHF1 are “involved in different processes, both regulating the signaling of molecules acting as neurotransmitters, namely acetylcholine and γ-aminobutyric acid (GABA). (…) This results in the regulation of GABA-mediated neurotransmission in the central nervous system, particularly in neocortical and hippocampal neurons” [8] (p. 12). An increased excitation by glutamate and reduced GABAergic inhibition [9,10,11] (inter al.) results in an overall excitation. Added to these, we have the long-distance innervation from monoamine-releasing neurons (noradrenaline, serotonin, dopamine), stimulating, inhibiting, or modulating the activity of the neurons they innervate.

Hyperexcitation of the primary visual cortex, for example, through glutamate, will result in more acute vision, as in autistic savants and normal subjects stimulated by transcranial magnetic stimulation (rTMS, below). Inhibition, excitation, and feedback loops are constrained in a different balance in the savant versus the normal brain, and by analogy, in the archaic versus the modern brain [12,13,14] (inter al.). The neural regulation and pathways for vision and language will stand central in our presentation. Bundles connecting modern language processing areas seem to have evolved within the main pathways from the visual areas to the prefrontal cortex; thus, visual stimuli may interfere more with the functioning of areas that are now commonly associated with language processing but also other tasks [15,16]. In addition, involvement through central areas is indicated [17,18].

## 2. Evidence, Analysis, and Systematic Review of the Research Literature

Experimentally, by using transcranial magnetic stimulation directed toward the left temporal lobe in normal functioning persons, it is demonstrated that a temporary halt in top-down cognitive control may improve abilities such as performance on visual memory tasks or numeracy tasks, lasting up to 45 min after repetitive transcranial magnetic stimulation (rTMS) [19]. rTMS as well as transcranial direct current stimulation (tDCS) [20] may temporarily knock out temporal areas for language. An obstruction to the activity in one area will often lead to improvement in other areas. Accordingly, rTMS and tDCS, reducing left frontotemporal and left anterior temporal lobe activity, may increase visual memory. In frontotemporal dementia, where language gradually becomes unavailable, there are many cases with strongly ameliorated drawing skills, even including examples of high artistic quality [21,22]. Interestingly, the oldest known orangutang (Molly) also started to paint prolifically at a later age [23], which may indicate that when some mental abilities degrade, others are liberated, and, also for other primates, higher cognition may block artistic expression.

Many people close their eyes when they focus on auditory sensations. Closing the eyes modulates attention [24], as the alpha wave activity increased significantly in comparison to opening the eyes in a dark room (i.e., the effect is not due to visual stimuli but rather from closing the eyes). This demonstrated “that the modulation of the human alpha rhythm by auditory attention is increased when participants close their eyes” [24] (abstract). We may also use vision when we learn the language, for detecting shared attention to a referent, for focusing on overt articulators, as in lip reading. Visually impaired children may therefore lag in communicative skills or show impaired language development [25,26]. This indicates that vision plays a role in language perception, as well as language development. Since we attend to visual cues for language, it might also mean that irrelevant information from the visual stream may interfere with language.

Interaction between vision and speech is demonstrated in the classic McGurk effect [27]. We all read lips, but when the lips conflict with the sound, the perceived sound changes in the direction of what we perceive visually. This is confirmed by Sætrevik [28], elaborating on the relationship between vision and speech, in a dichotic listening paradigm, with a video showing a mouth pronouncing a syllable (*ba, da*, or *ga*) that matched the left, right or neither ear. The results showed a significantly enhanced right ear advantage when the visual stimuli matched the right ear syllable, but when the visual stimuli matched the left ear, there was no significant difference in picking the sound of that ear. This shows an asymmetry for the McGurk effect that is consistent with vision interacting with speech perception in the left hemisphere (that has a right ear preference).

Vision also interacts with object naming in the so-called “bouba-kiki effect” [29]. Present a smoothly rounded amoeba-like drawing and a spiky-spiny form side by side and ask a person: Which of them is a bouba and which is a kiki? The overwhelming response is that the amoeba-like form is bouba and the spiky one is kiki. Ramachandran [30] explains this phenomenon by what he labels the “synesthetic bootstrapping theory”, according to which the visual and acoustic signals meet, resulting in a cross-modal abstraction. As to where the two perceptual inputs meet, “there are strong hints that the angular gyrus is involved in this remarkable ability we call cross-modal abstraction” [30] (p. 176ff). The angular gyrus, which is greatly expanded in humans, is involved in multimodal processing and can be considered a *rich club* node involved in both musical, visual, and language processing [31]. Interestingly, the effect was not detected in a primate (Kanzi) capable of performing highly accurate sound-to-word mapping [32]. Since bonobos do not articulate speech sounds, it is unexpected if there were a fast mapping between “round” and “sharp” articulation and the visual counterparts, but it is also the case that the angular gyrus is relatively less prominent in non-human primates, so the multimodal symbolism is less likely to occur spontaneously for that reason. The angular gyrus is one node where perception and cognition interconnect multimodally.

The mirror neuron hypothesis by Rizzolatti and Arbib [33] also indicates the existence of a crossroads between vision and the evolution of language. It regards “the mirror system for grasping as a key neural missing link between the abilities of our nonhuman ancestors 20 million years ago and modern human language, with manual gestures rather than a system for vocal communication providing the initial seed for this evolutionary process” [34] (abstract). This system is ultimately triggered by the visual brain when we see either an object or a hand grasping an object. This leads to the activation of two sets of neurons. Canonical neurons respond to a form seen by activation of a motor program, potentially leading to motor execution, such as hand/arm movement. Mirror neurons simulate the goal and the meaning of another person’s motor act, e.g., a hand gesture [34] (cf. their Figure 3, p.113). Hence, hand movement is significant in communication between our primate relatives, and it certainly became important in archaic Homo sapiens, as it still is in modern people’s communication. A high population of mirror neurons is found in Broca’s area (Brodmann areas 44/45) in modern humans, and in the Broca homologue in macaques, for example. Since visual and auditory information are found to converge in, or near, Broca’s area, the mirror neuron hypothesis of language evolution [33] has support in observations in both humans and our primate relatives, and possibly other mammals as well. The mirroring of a hand gesture will evolutionarily take steps toward speech [33], initially through vocalizations coordinated with a gesture to create redundant signals with the potential to set the hands free as vocal communication take over.

### 2.1. Neuroanatomy of Language Evolution

Marslen-Wilson and Bozic [35] discuss a dual neurobiological network in language evolution, stressing that a “strong evolutionary continuity between humans and their primate relatives is provided by a distributed, bi-hemispheric set of capacities that support the dynamic interpretation of multi-modal sensory inputs, most relevantly in the context of social communication between members of the same species” [35] (p. 176). Added to this archaic system is a left hemispheric frontotemporal network that is “significantly more extensive in the human than in even our closest primate relatives” (ibid). The two systems are evolutionary and functionally distinct but well integrated into daily language. The bi-hemispheric system can trace its neural substrate as far back as 25 million years ago with the appearance of macaques, for example. The left hemisphere frontotemporal system is far more associated with Homo sapiens [36] (p. 17443), [35] (p. 177). As we will see later in this article, there are many other areas involved in language processing, some of which are cross-modal, and some are less lateralized. We should also keep in mind that most of the brain areas do not have a specific dedicated language function but serve more general functions that can be used for language processing.

The left-lateralized perisylvian network, then connects the auditory cortex to Wernicke’s and Broca’s areas through the Arcuate Fasciculus (AF), a bundle of fibers that have been the subject of recent scientific discourses with respect to neuroanatomical differences between the brain of macaque and that of human, within the frontotemporal area. The data favors a “progressive evolution of the AF during primate evolution rather than great leaps that would indicate categorial changes of the AF’s morphology between contemporary species.” [37] (p. 86). Furthermore, “recent works in human and non-human primates suggest that the AF course and terminations are more similar between species than described before” [37] (p. 86).

The neuronal architecture of the connected language related areas was already present before the Homo lineage. Hage [38] concludes that in our primate relatives, there are two separate neural networks that co-operate during communicative vocalization, one cortical vocal articulatory motor network (Broca’s area (BA 44/45) in humans, Broca homologue (area F5) in monkeys and apes), and a sub-cortical network including the limbic areas and the motor cranial nerve nuclei, now used for innervation of the muscles for speech: “On the basis of comparative neurophysiological and anatomical studies, the proposed dual-network model reveals that distinct neuronal pre-adaptations essential for human speech evolution are also present in monkeys” [38] (conclusions). The evolution of language in a trans-primate perspective point to a continuous evolution from our primate relatives to at least the early stages of human speech [39], but these areas were, and to some extent still are, in the strident stream of information from primary visual areas on the way to the prefrontal cortex.

### 2.2. The Dorsal and Ventral Pathways for Vision and Language and Their Interaction

The visual pathways emerging from primary visual areas (V1/V2; Brodmann areas 17/18) running rostrally divide into a ventral and a dorsal stream. The ventral stream, also labelled the what-pathway, follows the inferior longitudinal fasciculus into the temporal lobe and the form recognition areas within the fusiform gyrus; moreover, the main color centers of the brain, areas V4 and V4α, are localized along the ventral stream. The dorsal stream, the where-pathway, follows the superior longitudinal fasciculus into the parietal cortex, continuing into the prefrontal areas. The where-pathway is color-blind but extremely sensitive to luminance contrasts, and this also relates to a balance of color-sensitive parvocellular and magnocellular cells, with the high temporal resolution, in the geniculocortical visual system [40] (p. 3417ff). It passes through the movement-sensitive area V5/MT. Moreover, this system is significant for our depth perception. The dual-route of the visual system was first described by Ungerleider and Mishkin [41] and later by Milner and Goodale [42], who redefined the dorsal pathway to primarily have a function in visuo-motor integration (hence, not restricted to a where-system) (cf. [43]) (p. 70).

A dorsal and ventral auditory pathway was later suggested by Rauschecker [44]. Regarding the dorsal pathway, fibers from the auditory cortex within the superior temporal gyrus (STG) run caudally to climb into the inferior parietal lobule (BA 40), then they turn rostrally, finally entering Broca’s area (BA 44, 45) through the superior longitudinal fasciculus (SLF) and through the more ventrally located arcuate fasciculus (AF). A branch also ascends dorsally, entering the premotor cortex (PMC, BA 6/8). The dorsal pathway brings phonological information from the posterior auditory cortex and the inferior parietal lobule (cf. [45] (their Figure 1, p.264), [46]).

The ventral pathway leaves the auditory cortex (BA 41, 42), runs rostrally within the STG, entering the extreme fasciculus (EF), ascending its fibers into BA 45 and 44 (cf. [45] (their Figure 1, p. 264)), where (in BA 44) they encounter fibers from the dorsal pathways. According to Hickok and Poeppel, the ventral stream “is involved in mapping sound onto meaning, (while) the dorsal stream is involved in mapping sound onto articulatory-based representations” [43] (p. 72), i.e., the dorsal stream has a sensory-motor function, connected to the muscles for speech. This is also mirrored in Levelt’s classic model of speech production [47] and spreading activation models for lemma activation [48,49]. There are also asymmetries in these streams. Hickok maintained that “(i)n contrast to the typical view that speech processing is mainly left hemisphere dependent, a wide range of evidence suggests that the ventral stream is bilaterally organized. The dorsal stream, on the other hand, is strongly left dominant” [50] (p. 2). The ventral stream is divided in three bundles: the inferior fronto-occipital fasciculus, the uncinate fasciculus, and the temporo-frontal extreme capsule. In a recent contribution, Weiller et al. [51] considered these bundles as one, as they pass through the extreme capsule, fanning out like a farfalle into respectively prefrontal and temporal cortices.

Rauschecker states that both the visual and the auditory “processing streams begin in early sensory areas and target prefrontal cortex as their ultimate endpoint” [45] (p. 265). This is similar, though not as lateralized, in other primates. Romanski outlines the anatomical features in macaques in detail relevant to the co-localization and intermingling of acoustic and visual information, as the ventral stream fibers from the inferotemporal cortex, carrying information about visual features, bend rostro-dorsally to enter the Broca homologue area of BA45. Auditory fibers from the STG ascend into BA12/47 anterior to this area [52]. This might be similar for both primates and other mammals, and it hints at the possibility for grand scale interaction between auditory and visual information in the homologues of the areas we now consider most relevant for language and implies the overwhelming of these areas with interfering visual information, which may effectively prevent the emergence of language in these species. The Broca homologue of a macaque is thus a meeting point of visual and auditory information. Visuospatial and auditory spatial fibers follow their respective dorsal streams as they run rostrally [52] (their Figure 1, p. i62). Bernstein and Liebenthal’s model of neural pathways for visual speech perception [53] in humans have both visual and auditory streams following dorsal and ventral pathways to the dorsolateral, respectively ventrolateral prefrontal cortices.

#### Interaction of Language and Vision: The Temporal Visual Speech Area

It is a mystery why it is so easy for most people to learn to read and write. This is a synesthetic in nature; when we see a written text, we perceive the spoken words in our minds, and thus it should be rare. However, if we think of language at the crossroads of visual and auditory, it makes sense that this interaction is natural and possibly predating language. Bernstein and Liebenthal indicated one special area as a visual speech area, the so-called Temporal Visual Speech Area (TVSA) [53] (cf. their Figure 1, p. 3). It is hard to conceive how such an area could have evolved for the function of literacy in the little time available since text became commonly available, so the area likely precedes literacy. Interestingly new evidence shows that auditory and visual stimuli are integrated. An fMRI study by Beauchamp et al. [54] found patches of integration of visual and auditory stimuli in the superior temporal sulcus. Vetter et al. [55] demonstrated that complex natural sounds and imagined sounds activate the early visual cortex in blindfolded participants. Moreover, it is suggested that the auditory perceived sounds and those that are mentally imagined may be mediated to primary visual areas by higher-level top-down multisensory areas. “Our results suggest that early visual cortex receives nonretinal input from other brain areas when it is generated by auditory perception and/or imagery, and this input carries common abstract information” [55] (summary). Their (ibid.) analysis has identified posterior superior temporal sulcus and precuneus as possible candidates for mediation of information from sounds and their imagery to early visual areas. “The precuneus has been identified as an area responding to both visual and auditory stimuli and possibly serving as an audiovisual convergence area” [55] (p. 1260). The evidence is building up, supporting a view that the structures we now use for language were always there at the crossroads between sensory information, visual and auditory. What seems to have happened is that the reliance on primary sensation has weakened, which has ameliorated the development of higher cognition. In addition, another area with multimodal function, the fusiform gyrus, which has sometimes been labeled a Visual Word Form Area, might be less lateralized and involved in many forms of multimodal recognition tasks, tasks that are involved in written word recognition but have not evolved for written word recognition [56].

### 2.3. Language Relevant Homologues in Primates

Brain architecture comparison of the prefrontal cortex shows a similar organization even between distant relatives, such as humans and macaques. However, these areas have developed tremendously [57] in relative size and convolution. Evidence from anatomical studies points to a coupling of the auditory and the visual pathways. The evolutionary reduction of brain volume in modern humans, particularly in the visual areas, leads to a more globule-like brain, a process that went on in parallel to domestication and neoteny (selection for gracile features). It is now evident from research on the macaque brain, as well as the human brain, that auditory pathways and pathways for speech [43]) and the visual pathways [41,42] follow a similar gross organization, i.e., that of dorsal and ventral tracts. The auditory/speech bundles and those carrying visual information interact in the Broca’s area homologue in macaques [52], and in humans in the superior temporal gyrus (STG) [53], which indicates that this interaction might have evolutionary reduced away from Broca’s area to set it free for other tasks. The STG is also implicated in autism [58]. Such models present anatomical correlates to how vision still aids our conception of language cf. [25,59], and how visual areas are activated to respond to auditory stimuli [55].

We want to suggest that the homolog capacity for language is still present in many non-human primates [35,37,39,60,61,62,63,64,65,66]. However, this language capacity is flooded by other processes, so overt language does not result in animals. This could be compared with the Cleaning of the Augean Stables: A flood of information accomplishes the task, hiding what is going on underneath. Here, the overwhelming river corresponds to the strong visual signaling through pathways that interact with homologues of language-relevant areas. The fertilizer in the stable corresponds to a capacity for language that might be flushed out by overwhelming perceptual information flows.

In a series of studies, Becker et al. [37,60,61,62] investigated language-relevant homologues in the Olive baboon (*Papio anubis*). First, the lateralization of the Planum Temporale (PT) is investigated in 35 newborn baboons, which found a significant leftward asymmetry present in the newborns, and this leftward asymmetry showed a significant increase when scanned later at 7 to 10 months. Grey matter asymmetry [60] shows that a left lateralization in infants is not unique to human infants. As the baboons clearly do not have language, it is hard to know the exact role of the asymmetric left lateralization, but it is likely in common with other primates. One possibility is that the primate brain is indeed prepared for communication (“language”), but something prevents this from happening. Perception, relevant in the here and now, may overwhelm these areas and effectively prevent recursive language-like.

Broca’s region and its homologues may not be fully identified in modern humans, and their ancestors since “(d)ifferences in microarchitecture have been identified between Broca’s region in the human brain and areas 44 and 45 as homologs of Broca’s region in ape and macaque brains ([64,65]) and hypothesized—together with other anatomical factors—to be responsible for the unique human ability of language” [63] (p. 2429). The alternative hypothesis is that perception, relevant in the here and now, may overwhelm the areas now involved in language and effectively prevent recursive language-like associations that go beyond the here and now.

Becker et al. [62] show an association between brain asymmetry and the individual use of communicative gestures in the Olive baboon (*Papio anubis*). This study (ibid.) provides a link between the observed brain asymmetries and gestural communication. The handedness in communicative gestures was clearly better predicted by the brain asymmetry than the handedness for the handling of objects. This suggests that brain asymmetries are related to communicative behavior. One common thought about the evolution of language in humans is that the pre-language was performed by gestures. The study (ibid.) shows a connection between brain lateralization and communicative gestures in a non-human primate. Obviously, modern baboons are not the precursors of humans, but it feels reasonable to assert that brain lateralization is not uniquely human, and likely such a process could have been present also in human precursors.

### 2.4. Self-Domestication

Shilton et al. [67] discuss the issue of self-domestication and self-control across social species. Domestication in animals and self-domestication in humans can be described as a selection process towards less aggression. Lower levels of aggression can be achieved through more self-control, hormonal regulation, and particularly serotonin regulation.

The meek shall inherit the Earth. Why is it that selection for less aggression within a species has a reproductive advantage? One possible answer is that self-domestication has its origins in the emerging hierarchical structures that we find in civilizations, where a minority of the population dominates but supports, in exchange for work or other favors, the majority of the population. Which members of the majority would likely be favored by the minority that controls resources? People who demonstrate less or more aggression? Possibly, humans might select partners that exhibit less aggression. Archaic males preserve robust features longer than females, who show earlier the gracile traits that are associated with self-domestication [68]. The pressure may thus not have been symmetric between the sexes. The offspring of the meek could have a reproductive advantage for many reasons. Thus meekness has a chance to accumulate simply based on the possibility of detecting the meek, the meek seeking protection, and the (re-)distribution of resources towards the meekest in the majority of the population. This does not necessarily imply sexual selection, as long as the meek are relatively rewarded for their meekness and that the reward affects the survival rate of the offspring. 

In a hypothetical scenario, a small difference, for example, between 1% and 5% growth, may accumulate quickly (1.01^100^ = 2.7; 1.05^100^ = 131.5). The disadvantaged of the dominated group might easily have an expected negative growth rate. In this scenario, we would not expect a reproductive advantage until the emergence of dominance hierarchies with a majority belonging to the dominated class (possibly consisting of slaves or simply people low in the social hierarchy). In more recent times, slavery was common, and there is no reason to assume that this was not also the case in archaic cultures, with the capabilities of war, conquering, and taking prisoners. “Slaves in Ancient Sparta outnumbered free individuals seven to one, according to the Greek historian Herodotus. […] In Roman times, there were slave markets in every city of the Empire, and wealthy families could have hundreds of slaves. […] The average life expectancy for slaves in Rome might not have been higher than seventeen years” [69] (p. 162), [70] (pp. 140–141). For the majority living under such conditions, meekness might have saved their lives beyond the expected short life if they were liked by their masters, who would be the selectors of meek features.

### 2.5. Signal Modification

The quality and intensity of neuronal signaling are achieved through neuroactive substances such as serotonin, the concentration which can be modulated. Serotonin modulation is associated with domestication [67]. Moreover, specific genes related to the *extracellular signal-related* (ERK) and *mitogen-activated protein kinase* (MAPK) pathways, stimulating, e.g., synaptogenesis, leads to stronger interconnections. This includes the glutamate receptors, which are significant in the domestication process: “… Together with the glutamate receptor changes […] modifications may have played an important role in generating aspects of the cognitive profile associated with modern humans, including a full-fledged language-ready brain” [71] (p. 11). Therefore, evolutionary selection can work on several levels with the effect that the balance between primary sensation and other types of neuronal activation can be altered, in line with our thesis. This does not exclude the possibility that neural circuits, over time, could also be altered.

### 2.6. Why Did Language Develop?

One attractive answer is that it emerged because it was already there, so it was inevitable once some constraints were removed. For the function of communication, it might even be easier to develop mind-reading than to develop language. In mind-reading, we would do what communicating computers do: we would send the information directly to the other, bypassing cognition. A computer could do it, so maybe we can? Maybe some of us do sometimes? Facial expressions reliably and accurately signal six or seven emotions [72], which are often interpreted empathically as if they were our emotions. We may feel sad when we see a sad face. Facial expressions can thus make us empathically feel the emotions, direct empathic signaling of a mental-emotional state, which may *bypass cognitive reasoning* [73], as in embodied simulation. 

So why did we not develop mind-reading or direct information transmission? The solution might be evolutionary. Keeping some information to yourself can have large benefits, and so can giving deceptive information in the right circumstances. The problem is, instead, how we can prevent sending an accurate version of our thoughts (having a skull may facilitate this). Language is a compromise between transmitting information, keeping information, distorting information, and judging information. In that complex situation, thought (and self-control) are even more important. Mind-reading would not stimulate thought as much, and it might not be flexible enough to develop symbols when we could just transmit exact information, possibly as multimodal sensory experiences. Individuals who transmitted their thoughts and intentions directly would likely be eliminated, rather than favored, by evolution.

One clue to the evolution of language is that the main brain areas that are involved in language also serve other functions [15,16] (such as working memory, music and motor skills). Several other areas, such as basal ganglia and feedback loops, may be involved in language processing, control and learning [15,17,18]). Many cortico-striatal loops of the basal ganglia innervate widespread areas of the cerebral cortex, including motor, cognitive and perceptual regions [17]. The information then enters closed and open loops [17]. “The presence of the visual cortico-striatal loop indicates that the striatum is able to interact with cortical regions responsible for sensory processing” [17] (p. 8). The picture that emerges is that there are no brain areas fully dedicated to language processing but rather a set of brain areas that perform functions that are relevant to language processing. Language is what happens when these areas are orchestrated in the right mix and adding themes stemming from previously processed information, while raw perception plays second fiddle. These other functions are mostly likely primary and existed in the homologues of these areas in primates and other mammals long before language existed. However, if these areas were wired such that they received input from extremely information-rich sources such as sensory areas, and vision in particular, their capacity for doing recursive processing on information that is far removed from direct sensation would be highly limited. 

Recursive processing is one hallmark of linguistic processing, and another is the ability to use information that is not present in the here and now. To achieve this, using the similar processes that previously dealt with a virtual flood of information puts a demand on making the sensory information less overwhelming. One feature of language is that we need someone to talk to, and we need a developed language for language to show its benefits. It takes modern children several years to acquire a fully developed external language, which indicates that the ability to process language must be present before the external language can develop.

### 2.7. Summary

The general schematic impression is that sensory information enters a neurological crossroads, situated in the vicinity of, or paths towards, the *classical language areas* and their homologues in other mammals, and their interconnections also with deeper structures, with many recursive feedback loops. The balance between the impact of sensory information and information that stem from recursive feedback loops, which may provide a filtered version of previous experience and distilled knowledge, is initially prioritizing the information that is valid here and now. We propose that a lowering of the intensity of sensory information will tilt the balance away from the here and now, which predicts that as this emerges, we will have an increasingly less accurate perception, and less accurate memory, as the details compete with information that we might label *“knowledge”.* This is obviously a disadvantage in the here and now—we might still experience details by attending to them, but this normally takes both time and effort. Being less observant of your surroundings is not obviously beneficial for the survival of an individual. However, becoming freer from the here and now may have other benefits and facilitate planning and coordination of larger groups. Therefore, this leap will likely not happen before the first steps towards culture and civilization have been taken.

## 3. Results

We have presented evidence for language-associated capacities in other primates. There is evidence of lateralization connected to communicative gestures in baboons [62]. There are also similarities in the anatomical layout of the homologues of language-associated areas (for example, homologues of Broca’s and Wernicke’s areas). Chimpanzees can carry out sequencing tasks with superhuman accuracy and speed [2], relying more on perception in the here and now, though the chimpanzees can be distracted for about 10 s and still be able to carry out the task[ibid.]. Humans, possibly as a consequence of having language, are able to focus their attention in order to perceive details, including fast-changing details. However, this focus may result in “inattentional blindness” [74]. When attending to something else, people may miss the unexpected gorilla banging his chest in full view.

One conundrum, if we assume that language emerged with early Homo sapiens 250,000 years ago, is what did we do with language until civilizations started to emerge 200,000 years later. Our solution is that the emergence of language may be associated with the reduced brain size in Homo sapiens that started about 50,000 years ago and more markedly 10,000 years ago [75,76,77]. More recent statistical analyses find an even more recent changepoint as close as 3000BP [78], which may be partly due to data selection and using a linear regression model. It takes time for trends to change and confidence intervals to narrow. The shrinking brain is recent; we find the evidence consonant with a gradual shrinking that may have started approximately at the end of the Upper Paleolithic and accelerated even more recently as changes rippled through the human population. Apparently, a smaller brain was beneficial for our species.

The reduction in brain size is also associated with the emergence of agriculture and larger group size, and the loss is often explained as a result of self-domestication. We suggest that an effect of this loss in brain size was the reduction of neuronal signaling and/or pathways related to raw perception and vision in particular. Visual perception relies on informational highways that may provide so much information that it can be overwhelming for other brain functions, such as retrieving knowledge appropriate to the situation or imagining something that is not present in the here and now. We hypothesize that the loss in brain volume is mainly linked to reduced perception of detail in space and time. We are no longer able to perceive how many hooves of a running horse touch the ground, as the cave artists of Chauvet may have seen with ease [1]. Modern people often reduce these details to a schematic rhythmic pattern—clip-clop—instead. However, there are still visual thinkers around, and their brains can have the broad highways that allow them to perceive tremendous detail, particularly in movement.

We hypothesize that the evolution of language followed the decline in visual acuity at the end of the Paleolithic when we no longer find evidence for elaborate realistic cave paintings (although we find iconic and symbolic cave paintings after this period). The presence of the gene FOXP2, a marker for speech and language abilities, coincides with this interesting period. “The dating of the FOXP2 gene, which governs the embryonic development of (the necessary) subcortical structures, provides an insight into the evolution of speech and language. The starting points for human speech and language were perhaps walking and running. However, fully human speech anatomy first appears in the fossil record in the Upper Paleolithic (about 50,000 years ago) and is absent in both Neanderthals and earlier humans” [79] (abstract). However, the crucial factor for language is not a fully developed speech apparatus but possessing a language-capable brain.

The loss of realistic cave painting and the development of language may thus coincide in the Upper Paleolithic, approximately 50,000 to 10,000 years ago, a period when we also see a significant loss of brain volume. The pattern of allele frequencies among humans indicates that a functional change occurred that could be responsible for a positive selective sweep affecting the FOXP2 gene in the last 50,000 years [80,81]. Thus, it is possible that the emergence of language happened recently during the evolution of fully modern humans.

The precursors of language, as we argue, are much older. A study on the hominid remains of the external and middle ear of more than 500 thousand-year-old fossils from the Sima de los Huesos site (Sierra de Atapuerca, Spain) “show that these hominins had the same auditory capacities as modern humans” [82] (abstract). The modern FOXP2 gene had archaic variants, which may indicate language precursors present in both archaic Homo sapiens and neandertalensis. “Interestingly, the recent discovery that Neanderthals share with modern humans two derived substitutions in the FOXP2 gene offers tantalizing new evidence for the possible presence of language in Neanderthals. At the same time, the beginnings of human speech have recently been suggested to date to the origin of the genus Homo” [82] (p. 4180), cf. [83]. The effects of FOXP2 variants have lately been observed in animal models, as at least “two amino acid substitutions in the FOXP2 protein that are shared with archaic humans [83,84,85] affect synaptic plasticity and dendritic trees in cortico-basal ganglia circuits when introduced into mice” [80] (p. 849).

We are not suggesting that any singular mutation in FOXP2, for example, is responsible for language, but we rather suggest that *downregulation of perceptual information* may have created the conditions for language to emerge, using older functions now able to access more information outside of the immediate here and now. A recent study has shown downregulation of dopamine in the brain, as well as increased dendritic length and long-term synaptic depression when FOXP2 was introduced in genetically modified mice [86]. Other studies have shown that depressed levels of dopamine affect vision negatively [87,88]. However, FOXP2 might be involved in downregulating dopamine.

Gould and Lewontin [89] and Gould and Vrba [90] introduced the term *exaptation* for the evolutionary shift in function. Previously, this was thought of as a pre-adaptation, i.e., somehow nature was preparing for something that would happen much later. Exaptation suggests that a previously important function can get a different function in a changing context; feathers may have started as an adaptation for conservation of body heat and later facilitated the emergence of flight in birds. This article suggests a similar process to exaptation, but we focus on the (possibly gradual) removal of a useful feature that was an obstacle to the development of other abilities, the liberation of potential in other structures. One example might be how losing a tail may have made walking more efficient, even though the fact that having a tail helps to balance, therefore losing the tail would demand more brain control, just as modern fighter jets are structurally unstable and controlled by computers.

We might need a term for an evolutionary fading out or receding to leave room for something else that is already rudimentary present; possibly, the process could be called recidation. The process suggested in this article is likely mediated through neuromodulation, neuro-signaling, and neuronal growth. Thus, it is difficult to observe this directly in the fossil record as these tissues rarely, if ever, fossilize. However, there have been tremendous advances recently in reconstructing archaic genomes that may make the changes observable in the near future.

## 4. Discussion

The main theme of this article is to suggest that the full language capacity emerged much later than the emergence of Homo sapiens and that this coincides with the shrinking brain in the Upper Paleolithic (50,000 to 10,000 BP) and the eventual disappearance of highly realistic cave paintings, which were replaced by iconic and symbolic cave art. The article has laid out some evidence, which is attractive for understanding the evolution of language. With the much more recent advent of language, there is less of a need to explain why we do not see a more rapid development of the human condition for over 200,000 years, as language would have allowed the communication of imaginary or hypothetical scenarios and untied our minds from the here and now. We have indicated that vision and language still interact, and this may be a clue to why it is relatively easy for modern humans to acquire reading and writing. There is no time after the event of writing for realistically evolving the synesthetic requisite of activating an internal voice when seeing a visual encoding of spoken language. One solution is to rely on the much older interplay between different modalities. Similarly, we use visual encodings for transcribing music and instruct our hands to perform using instruments. Motor pathways are often co-activated with Broca’s area in language studies, possibly because of covert rehearsal [15].

During our explorations, we touched on the nature of the necessary change for language to emerge. Recent research in transcranial stimulation [19,20,91,92], but also studies of Frontotemporal dementia (FTP) [93], led us to believe that knocking out some dominant, overwhelming neural signaling could set other pre-existing abilities free. This convinced us that it is possible that the brain’s capacity for processing complex recursive language could have been set free by lowering the intensity of neuronal information related to raw perception and raw visual perception in particular. Several converging pieces of evidence suggest that the timing of a fully developed language apparatus coincides with a shrinking brain and a loss of superior processing of sensory information [1]. Interestingly, the drawing ability of primates may also improve when their higher cognition deteriorates [23]. Thus, knocking out the brain areas (e.g., Broca’s homologue, prefrontal areas) related to higher cognition may improve drawing abilities in primates and show that interference between raw perception and higher cognition is present also in other primates.

One prediction from our hypothesis is that the pre-cursors of language were, and are still, present in other animals. This capacity is prevented from emerging by the useful but overwhelming processing of sensory information. The pre-cursors of language may simply be more general functions that are too busy processing perceptual information in the here and now. If we could lower or alter the flow of information-rich sensory information, it would mean that animals could perform better on language-like tasks, for example, demonstrating an improvement in understanding more complex commands. Similarly, we do not claim that modern humans are at the absolute peak, and thus also, human performance can be enhanced by altering the neuronal flow of information, as in transcranial stimulation studies. This could happen by psycho-active drugs, transcranial stimulation, or sensory deprivation. The experiments are deemed impractical, as it would take tremendous time to train the animals and possibly several generations to get reliable results.

However, there are already animals that can perform impressive language-related tasks. Chimpanzees and gorillas have been taught to use a rudimentary form of sign language (as it is virtually impossible for them to coordinate their vocalization to produce anything close to speech). The sign language of primates may not be fully linguistic, but it does extend to the capabilities of communicative gestures that are natural to these animals [94,95]. Alex, a grey African parrot, was taught how to do tasks such as picking the correct object and performing some rudimentary arithmetic [96,97]. Rico, a border collie, demonstrated reasoning by exclusion, an ability to pick the correct novel toy when told to fetch a toy with a novel name [98]. Several species, domesticated species, in particular, have the ability to reason by exclusion [99], an ability that not long ago was thought of as a uniquely human capacity. 

Lately, studies [62] (inter al.) have shown brain lateralization related to communicative gestures in the Olive baboon, and likely such lateralization is present in other primates as well, but it is still unknown if similar lateralization is present in the above-mentioned wide variety of species (primates, birds, dogs). If language evolution is gradual, as the most recent studies indicate, it is plausible that the general mechanisms that are later used for language have been present for possibly millions of years. The reason that only humans have fully developed language is that something blocked the path towards a brain that is not busy with the hurly-burly of living in the moment. Possessing language may also be related to our ability to focus on details, which may cause us to experience blindness to sensations that are not in the focus of our attention [74]. That this liberation of mind is so rare is possibly because this obstacle (effortless, detailed, accurate and fast perception of the present) is highly useful for survival. The present is where it happens. Once that obstacle has been lowered, the ability to efficiently communicate about the past and the future, as well as about the present, creates a different evolutionary scenario, where it is possible to move forward even when losing brain volume associated with accurate sensory perception. That language interacts with vision (e.g., multimodal perception) indicates that they draw on similar resources. We argue that perception, and vision, in particular, remained a strong priority until recently, which blocked full language abilities.

This article has outlined converging evidence for the more recent emergence of language, relying on pre-existing functions that were available much earlier. We place the emergence of full language ability at 50,000 to 10,000 years BP. As the emergence of Homo sapiens is much earlier, and has moved increasingly earlier [100], it is a conundrum as to what we did with language for so long *if* language appeared simultaneously with our species. We are sympathetic with the idea that there are no brain areas fully dedicated to language. Furthermore, the many areas that serve language-relevant functions are present in non-human primates and possibly other animals as well. Language and higher cognition compete with raw perception in the here and now. Higher functions are facilitated when raw perception can be delegated to the background, which necessitates focused attention for perceiving details, as attested in inattentional blindness. This article has indicated a few rich club nodes where information from both vision and language meet, for example, the Angular Gyrus and the Superior Temporal Gyrus. Detecting genetic changes in the regulation of signal substances via serotonin and glutamate receptors in the ERK/MAPK pathways is a starting point and is restricted by onset in the Upper Paleolithic. This view opens many other avenues for testing the hypothesis also in modern humans. For example, temporarily knocking out specific brain areas related to language or higher cognition may show increased perceptual abilities and vice versa. This could be relevant in experiments on both human and non-human primates. A recent emergence of full language also facilitates the mapping of relevant genetic changes, as we are looking at a defined time period.

## Data Availability

This study is based on literature review and analysis.

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
