# Peer review of "Is Reduced Visual Processing the Price of Language?"

_brainsci, 2022, doi:10.3390/brainsci12060771_

Round 1

Reviewer 1 Report

I enjoyed reading this paper focusing on the question: “Is reduced visual processing the price of language?”. Overall, I found it well written and quite interesting. That said, I do not believe that the main sections of the paper are correctly well-organized. I mean, it is very difficult to categorize an entire section of the paper as “Materials and methods”. This looks much more like a discussion of related topics. Indeed, there are no materials nor methods to present in this paper. Similarly, also the Results’ section is not really a Result section. I suggest the authors to remove these labels. I also suggest the authors to further answer to the following points: 

1.     Page 1 – Introduction: 4 to 6 lines from bottom: “the linguistic abilities of visual thinkers may be impaired, which suggests a negative correlation between visual perception/memory and language. Possibly, this negative correlation involves an impaired propensity for symbolic thinking”: Please provide references for this claim. 

2.     Page 2 – Introduction: 3-4 lines from above: “Compared to cave painters, modern man has generally lost the extreme ability to remember visual details and reproduce them artistically”: This claim is way too strong. Think about artists. I would put things this way. Please, moderate such claims.

3.     Page 4 paragraph 2.2 1st line: “discusses” should read discuss

4.     Pages 4 & 5 par. 2.2: while describing the dual neurobiological network in language evolution, referring to Marslen-Wilson & Bozic the authors describe the presence of a left-lateralized fronto-temporal network involved in language and a not-well specified “distributed bi-hemispheric network” implicated in the interpretation of multimodal sensory inputs needed for social communication. I suggest that the authors provide more info about this second network. What areas are involved in it? Could they provide also some additional references? Furthermore, I suggest that the authors moderate the claim about the presence of a left-lateralized network for language. Indeed, growing evidence suggests that what is mostly left-lateralized is the network for lexical processing and not for language. Language is simply much more than just lexical skills. Furthermore, even lexical abilities are implemented in a wide range of neural networks encompassing left and right cortical and subcortical areas (even epicenters in the right cerebellar hemisphere). Not to mention the cases of crossed aphasia supporting the idea that at least some individuals have a mostly right-lateralized network for lexical production and comprehension. All in all, I suggest the authors to moderate the claim and rephrase the sentence as follows: “a mostly left-lateralized fronto-temporal network involved in lexical processing”

5.     Page 5 paragraph 2.3 5th line: “is localized” should read “are localized”

6.     Page 6 paragraph 2.3.1 6th line: Do the authors refer to the Visual Word Form Area? If this is the case, why don’t they mention that this is in the right fusiform gyrus and that it is an area originally bilaterally involved in the ability to organize specific visual traits in specific configurations (e.g., face area in the right hemisphere)? I would like to see here also a consideration about this point. This may be a further piece in the puzzle that the authors are trying to build up. 

7.     Page 7 paragraph 2.3.1 1st line: “the structures we now use for language is…” should read: “the structures we now use for language are…”

Author Response

Thank you very much for such kind words. We have revised and correct all the suggestions.
Comment 4: Marslen-Wilson & Bozic is about the continuity of processing from ancestral primates to modern humans. Lateralization is important, but modern non-human primates also show lateralization for communicative gestures, as we come to later in discussing Becker et al.'s findings. The left lateralization is not only lexical processing, and as we pointed out the language associated areas perform many other tasks.

"Added to this archaic system is a left hemispheric frontotemporal network that is “significantly more extensive in the human than in even our closest primate relatives” (ibid.). The two systems are evolutionary and functionally distinct, but well integrated in daily language."

Comment 6: it is not the Visual Word Form Area, but we have added this with a reference to the discussion. It points to one more area where a later function (identifying a written word) rely on much older functions and interplay between modalities.
The TVSA: see (link in Bernstein & Liebenthal):

We added a reference to VWFA "The evidence is building up supporting a view that the structures we now use for language were always there in the crossroads between sensory information, visual and auditory. What seems to have happened is that the reliance on primary sensation has weakened, which has ameliorated the development of higher cognition. In addition, another area with multimodal function, the fusiform gyrus, which has sometimes been labeled a Visual Word Form Area, might be less lateralized and involved in many forms of multimodal recognition tasks, tasks that are involved in written word recognition but have not evolved for written word recognition[55]."

Reviewer 2 Report

It was one of the most interesting articles I have ever reviewed. Unfortunately, I cannot assess the relevance of information related to neuroanatomy and other "biological" issues. From the linguistic point of view, though, the work is tantalizing as it proposes the existence of negative correlation between reduced visual processing and the evolution of language, which is novel to me. 

It is nonetheless quite difficult to test the authors hypothesis. I also do not understand the step-by-step methodology the authors applied to get their results.

On the other hand, I have no specific comments to the English language, spelling, or formatting. There are no figures or other visual aids in the article.

Abstract made me want to learn more! Introduction logically builds up to the research question. Concerning the content, I am not sure if the title of chapter 2, namely Materials and Methods, is the best choice, though. The authors collect data supporting their theory from different fields, spanning a wide scope of areas, which suggests a holistic approach to the topic, including hints to neuroanatomy, history, biology, sociology, art, and more.

In Results and Discussion the authors hypothesize that a smaller brain was beneficial for homo sapiens. They argue that human language as we know it, may have emerged between 50,000 to 10,000 years ago. In other words, much later than our species appeared on the face of Earth. Moreover, it could happen only when a shrinking (!) human brain lost its superior ability to process sensory information, thus allowing higher cognition to take its place - in an evolutionary shift called by the authors "exaptation" (a previously important function gives place to a new one). 

Author Response

Thank you very much for your kind and encouraging words.
The review summarizes the main points accurately. We have made some revisions. One is to point out that language acquisition works best in a community of speakers of the language, which points to culture preceding language.

end of paragraph 2.7
"To achieve this, using the similar processes that previously dealt with a virtual flood of information, puts a demand on making the sensory information less overwhelming. One feature of language is that we need someone to talk to, and we need a developed language for language to show its benefits. It takes modern children several years to acquire a fully developed external language, which indicates that the ability to process language must be present before the external language can develop."